# Effect of Outdoor Sports Participants on Leisure Facilitation, Recreation Specialization, and Leisure Satisfaction: Yacht and Golf Participants

**DOI:** 10.3390/ijerph18158128

**Published:** 2021-07-31

**Authors:** Young-Hoon Kwon, Young-Kyu Cheung, Byoung-Wook Ahn

**Affiliations:** 1Department of Leisure Marine Sports, School of Marine-Sports, Seosan Campus, Hanseo University, Seosan 31962, Korea; marine-boy1@naver.com; 2Hanseo Marine Sport Training Center, Department of Education, Taean Campus, Hanseo University, Seosan 32158, Korea

**Keywords:** leisure facilitation, recreation specialization, leisure satisfaction, outdoor sports

## Abstract

The purpose of this study was to investigate leisure satisfaction among outdoor sports participants in golf and yachting. Influence was also measured of recreation specialization on leisure satisfaction, and the effect of the relationship between leisure facilitation and leisure satisfaction on golf and yacht participation was investigated as well. Frequency, reliability, confirmatory, and correlation analysis, as well as structural equation modeling results, indicate that leisure facilitation had no influence on outdoor sports participants’ recreation specialization. Leisure facilitation had a positive influence on leisure satisfaction among the golf and yachting participants, and their recreation specialization had a positive influence on their leisure satisfaction. The effects of the COVID-19 pandemic are addressed, specifically the constraints that the disease has imposed on outdoor sports and leisure, and strategies are presented for addressing these constraints and promoting outdoor sports participation.

## 1. Introduction

Modern societies have achieved a high degree of industrial growth attributable to technological advances. These changes have brought material abundance, and the introduction of a 40-h work week improved quality of life and increased leisure time and participation in various sports activities [1]. The growth of the outdoor sports market is due to the continuous expansion of numbers of participants [2]. Golfing and yachting are representative untact outdoor sports that share the advantages of using a wide outdoor space and allowing for participation by individuals or small groups, and demand for and participation in both are increasing steadily.

Following the recent phenomenon of global urbanization, modern people are pursuing nature-based outdoor sports to relieve stressors of urban life such as noise, congestion, and air pollution [3]. According to Statista [4], the world’s largest outdoor sports market, the United States is worth $36.3 billion, followed by China’s market at $20.5 billion, and the United Kingdom’s at $3.3 billion, with Korea’s market coming in fourth place at $2.1 billion. Given the changes that have taken place already and those predicted for the future, it is necessary to analyze the impacts of COVID-19 on outdoor sports and to respond to the changing sports and leisure culture. As of September 2020, COVID-19 had caused sales declines in 7.4% of sports facility businesses, 2.3% of sports equipment businesses, and 2.4% of sports service businesses [5]. The number of companies that closed due to cost burdens and other factors increased dramatically.

However, although the global pandemic brought confusion throughout societies worldwide, in Korea, it was an opportunity to promote outdoor sports and recreation; participation increased in activities such as nature tourism and individual and small-scale untact sports. The outdoor sports and leisure industry is now expected to grow rapidly in line with changes in demand. In particular, golfing and yachting are representative outdoor untact sports that are attracting attention but have not been the subjects of much exploratory research.

The aim of this study was to analyze the effects of leisure facilitation, recreation specialization, and leisure satisfaction on outdoor sports participation and to verify any structural relationships. Raymore [6] defined leisure facilitation as referring to factors that form or promote leisure preferences and strengthen or encourage participation in leisure. Bryan [7] first defined recreation specialization as a continuum ranging from engagement by the general public to specialization by experts and observed that it can be measured in various ways. Sa [8] found that, among subfactors of leisure facilitation that influenced recreation specialization, intrinsic, interpersonal, and structural factors had significant effects. Leisure facilitation premised on continuous participation is likely to have strong a correlation with recreational specialization, which focuses on the accumulation of expertise. Indeed, a number of researchers have confirmed such a correlation between leisure facilitation and recreational specialization [9,10,11]. Kim [11], in particular, identified this positive relationship among ski participants. Based on the above theoretical background, we proposed the following initial study hypothesis:

**Hypothesis** **1.**
*Leisure facilitation of outdoor sports participants will have a positive effect on recreational specialization.*


Beard and Ragheb [12] first defined leisure satisfaction as a positive awareness or emotion formed, elicited, and obtained as a result of participation in leisure activities and opportunities that results in the satisfaction of personal needs. In one previous study, Lee [13] found that facilitation of daily leisure among college students had a positive effect on all factors of leisure satisfaction; the author also analyzed effects of the relationship between leisure satisfaction and college life satisfaction, and found that psychological and environmental leisure satisfaction had direct positive impacts on college life satisfaction. Hong, Seo, and Lee’s [14] research result that the desire for leisure facilitation itself affected leisure satisfaction can be said to support the theoretical background of this study. Kim [15], as well, found that leisure satisfaction tended to increase significantly as the frequency of and length of participation in leisure activities increased. In contrast, leisure facilitation of marine sports participation did not affect participants’ leisure satisfaction. According to a study by Ahn and Kwon [16], the purposes of existing marine sports remain simple experience, skill acquisition, and ability improvement, and leisure satisfaction was affected by the lack of emotional leisure of relaxation from healing scenery. Based on these theoretical findings, we proposed the second study hypothesis:

**Hypothesis** **2.**
*Leisure facilitation of outdoor sports participants will have a positive effect on leisure satisfaction.*


A number of domestic researchers in Korea have addressed recreational specialization and leisure satisfaction. For instance, Jeong [17] determined that recreational specialization can have positive effects on personal aspects such as quality of life, leisure satisfaction, and self-development, and Jeong and Hwang [18] interestingly found that low recreational specialization combined with longer participation had the strongest positive effects on leisure satisfaction. However, Lee and Hwang [19] also found significant positive effects of skill level, importance, and technical knowledge of recreational specialization on leisure satisfaction; among the recreational specialization subfactors, the authors found that skill in the behavioral domain had the greatest effect on leisure satisfaction: the more skilled activity participants perceived themselves to be, the greater their satisfaction with their leisure activities. Lee and Lee [20] found that the recreational expertise of women participating in rhythmic exercise had a positive effect on leisure satisfaction; specifically, importance as a sub-variable of recreational specialization had positive effects on psychological, educational, social, restful, physiological, and environmental satisfaction, which are all sub-variables of leisure satisfaction. Gu [21], meanwhile, analyzed leisure satisfaction among a group of Chinese walking tourists centered on levels of recreational specialization, and those findings also support the theoretical background of this study. Based on the above findings, we proposed the following final study hypothesis:

**Hypothesis** **3.**
*Recreation specialization of outdoor sports participants will have a positive effect on leisure satisfaction.*


The above literature findings highlight that active research is ongoing in leisure facilitation, recreation specialization, and leisure satisfaction both domestically in Korea and internationally, and the findings suggests certain structural relationships. However, even though outdoor sports participation is quite in the spotlight rather than stagnating amid the current strict pandemic-related restrictions on and for leisure, research on these relationships remains conceptual. Accordingly, we determined it necessary to study the structural relationships among leisure facilitation and recreation specialization, leisure facilitation and leisure satisfaction, and recreation specialization and leisure satisfaction.

With this study, we sought to facilitate leisure, recreational specialization, and leisure satisfaction in participants in two outdoor sports, golfing and yachting. Currently, COVID-19 has led to many restrictions on participation in leisure activities, however, participation in outdoor sports is recognized as increasing vitality and immunity in our lives. Thus, we aimed to analyze the psychological factors of leisure among outdoor sports participants and provide data to support policymaking toward developing outdoor sports.

## 2. Materials and Methods

### 2.1. Instruments

For this study, we measured leisure facilitation, recreation specialization, and leisure satisfaction using preexisting survey questions. First, we measured leisure facilitation with scale items from Raymore [6] and items adapted from Kim and Lee [22] to suit this study’s purpose. Respondents were to rate all leisure facilitation items on 5-point Likert scales ranging from 1 (very unlikely) to 5 (very agreeable). Example questions include “my personality is extrovert” (intrinsic) and “my leisure activities are supported by people around me” (interpersonal), and participants gave information on their leisure activities as well (structural).

Second, we measured recreation specialization by revising items from the scale developed by Lee et al. [23] to meet the purpose of this study. Respondents rated these items on 5-point Likert scales that ranged from 1 (very little) to 5 (very much). Items included “knowledge of simulation golf” (cognitive), “participation in simulation golf for a long time” (behavioral), and “enjoyment of simulation golf in particular” (emotional).

Third, we measured leisure satisfaction using a scale developed by Ahn [24] for Korean adults. Respondents rated these items on 5-point Likert scales that ranged from 1 (very little) to 5 (very much). Example items include “enjoyment and stress relief” (stress solution), “building health and confidence” (health promotion), “continuous participation because of the need for skill” (technical development), “kind and understanding with each other” (interpersonal relations), and “self-development and happy time” (self-development).

### 2.2. Data Processing and Analysis

We used SPSS 21.0 and AMOS 18.0 for the data processing in this study. We used SPSS 21.0 to analyze the frequencies of sociodemographic characteristics and the reliabilities and correlations of the subject variables to verify the relationships among leisure facilitation, recreation specialization, and leisure satisfaction. We analyzed the structural equation model for confirmatory factor analysis and tested the study hypotheses using AMOS 18.0.

### 2.3. Ethical Clearance

All study procedures were reviewed and approved by the Hanseo University Department of Sports Research Institutional Review Board and conducted according to the principles expressed in the Declaration of Helsinki. After we explained the purposes and length of this research study, respondents provided consent to participate in this study. All participants agreed to allow researchers to use their personal information obtained from questionnaires for the purposes of this study, and all understood that they could refuse to continue to participate in the study at any time.

## 3. Results

### 3.1. Participants

The subjects of this study were adult men and women who were living in Seoul, Incheon, and Gyeonggi-do, Korea, and had participated in golf and yachting activities for more than one year. We used convenience sampling, a non-probability sampling method, to recruit the study participants; convenience sampling refers to randomly selecting a sample according to the convenience of the researcher. For this study, we limited subjects to golf and yachting participants, distributing a total of 500 copies of the questionnaire. We excluded 69 that we judged to be duplicates or that had responses omitted, leaving 431 completed surveys for empirical analysis.

Regarding the demographic and sociological variables of the study subjects, there were 228 men (59.9%), and the largest proportion, 32.0% (138 survey respondents), were in their 50s. Over half of the respondents, 248 (57.5%), were golfers, and the largest proportion, 29.0% (125 respondents), had participated in golf or yachting for more than one year but less than three. Regarding their intensity of participation, 152 respondents (35.3%) engaged in their activity for less than one hour per session. Table 1 presents the detailed survey respondent characteristics.

### 3.2. Validity and Reliability

We validated the research tool for this study using confirmatory factor analysis (CFA). CFA is a procedure for confirming inherent factor dimensions and hypotheses based on researcher knowledge [25]. The fitness indices we used were χ^2^/df, comparative fit index (CFI), the Tucker–Lewis index (TLI), and root mean square error of approximation (RMSEA). First, Carmines and McIver [26] proposed the standard χ^2^/df fitness standard, and it was 3.0 in this study. Bentler [27] proposed the CFI, and Bentler and Bonett [28] proposed the TLI, and adequate fit is represented for both by values of 0.90 or higher. The RMSEA was published by Steiger and Lind [29], and the fit criterion was set at 0.08 or less [30]. Table 2 presents the CFA results, indicating that χ^2^/df, CFI, TLI, and RMSEA all demonstrated adequate model fit. We calculated Cronbach’s alpha coefficients to verify the reliability of the survey instrument for measuring leisure facilitation, recreation specialization, and leisure satisfaction. Table 3 presents these findings.

### 3.3. Correlations of Study Variables

We calculated the correlations among leisure facilitation, recreation specialization, and leisure satisfaction among participants in golf and yachting as outdoor sports, and Table 4 presents the findings. Notably, leisure facilitation showed no correlation with recreation specialization. In addition, there was a positive correlation between recreation specialization and leisure satisfaction (Table 4).

### 3.4. Results of the Study Model

To elucidate the relationships among leisure facilitation, recreation specialization, and leisure satisfaction of outdoor sports (golf and yachting) participants, we tested a structural equation model of the hypothesis. Table 5 presents the detailed findings including the model fit indices. Briefly, Hypothesis 1, that leisure facilitation would have a significant effect on recreation specialization, was rejected. Hypothesis 2 of a significant effect of leisure facilitation on leisure satisfaction was supported, and Hypothesis 3 of a significant effect of recreation specialization on leisure satisfaction was also supported.

## 4. Discussion

Despite the constraints on all sports and leisure owing to the COVID-19 era and with the resulting spread of untact culture, outdoor sports such as golf and yachting are in the spotlight today. For the current study, we investigated the correlations among leisure facilitation, leisure satisfaction, and recreational specialization among participants in these two sports who were residing in Seoul, Gyeonggi, and Incheon, Korea. We discuss the results based on comparison and analysis with previous study findings.

First, leisure facilitation did not have a positive effect on recreation specialization among the outdoor sports participants in this study. Lee [9] found that the intrinsic, interpersonal, and structural facilitation of leisure facilitation had positive effects on recreational specialization and argued that past experiences with one’s own desired leisure and confidence from those experiences led to more positive perceptions of leisure activities and more active participation in them. However, these findings of a close relationship between leisure facilitation and recreation specialization were contrary to the results here. We expect that the inconsistency is because earlier findings do not reflect the peculiar impacts of COVID-19 on leisure facilitation factors; in this study, changes in structural facilitation such as cost burdens from increased opportunities and the expansion of untact culture and changes in factors of outdoor sports participants, such as intrinsic facilitation, did not have positive effects on recreational specialization. Park [31] emphasized the increase in non-face-to-face exercise regimens such as home training and online exercise that resulted from the prolonged COVID-19 pandemic and suggested an “instant group athletic culture” that aimed to improve accessibility and “loose solidarity” through new platform proposals.

In other research, Im [32] found that yachting participants were immersed in their activities through serious leisure experiences but that there were also constraints or elements of conflict; the primary challenge now was that cruise yacht sailing is not an activity one can participate in alone, and there are time limits on activities that engage more than five people. Specifically, effective 4 January 2021, the Ministry of Public Administration and Security [33] expanded nationwide the ban on private gatherings of five or more people that had been in effect only in the metropolitan area since 23 December 2020. The ban and other measures remain in effect to date, and these measures have constrained nationwide yachting competitions. In Im’s study, yachting competitors said that, in preparation, they were “changing my schedule for sailing,” “working to keep the sailing schedule,” and conducting “preliminary exploration for the game.”

The economic factors of yachting have a static effect on whether the leisure facilitation of yachting leads to recreational specialization. However, because of COVID-19, the yachting competition in 2020 did not proceed normally: the Korea Yacht Association website indicates that 8 of the 12 official 2020 tournaments were cancelled, among them the Busan Super Cup International Yacht Tournament, Admiral Yi Sun-sin’s International Yacht Tournament, the South Coast Cup International Yacht Tournament, and the Laser Radial National Championship [34]. Such cancellations have removed the reason for training and acquiring professional skills in yachting, competition, which has been the primary reason for the decline in recreation specialization. Competing in yachting gives a sense of accomplishment from having immersed oneself in overcoming intense situations that require high-level skills, and removing competition might naturally have decreased the motivation to pursue those skills.

Kim [35] researched golf participants and argued the need for virtuous circles of golf courses, users, and workers for safe enjoyment of the sport during the COVID-19 pandemic to overcome the challenges to both participating in sports and operating sports-related businesses. Choi and Kim [36] reported that financial stress was a negative impact of the psychological stress relief from golfing because higher frequency of golf participation poses a considerable cost burden, but higher-frequency participation increases the desire to participate and achieve success in order to maintain health. In short, there was likely some impact of coronavirus-related cost factors in the relationship between leisure facilitation and recreational specialization in both golfing and yachting.

Second, we found that golfing and yachting leisure facilitation had positive impacts on participants’ leisure satisfaction. Song and Yeo defined leisure satisfaction as a subjective emotion that is determined by individual expectations and satisfaction [37]. Song reported that the positive emotions felt while participating in leisure activities played a role in enhancing life satisfaction through leisure facilitation [38]. We judged this to be the reason that demand for and participation in outdoor sports are increasing; these sports facilitate personal or individual participation in outdoor leisure activities even as other types of recreation, such as group and indoor sports, are unavailable. Kim, Seok, and Im [39] observed that sports activities enable continuous training and body management and that changes in physical abilities and appearance through exercise enhanced participants’ leisure satisfaction, and this is one pathway whereby leisure facilitation might influence leisure satisfaction. In particular, golfing and yachting, which allowed for participation by small numbers of people using the natural environment, satisfy the social, personal, and structural leisure promotion requirements of untact culture. Both were increasingly being recognized for their value as future-oriented sports.

Researchers have established sailing as an attractive sport that offers both beauty and education and that gives pleasure from experiences and elements of competition; for instance, Im recognized yachting as a process of “traveling” that offers a variety of satisfactions [32]. In other words, active participation in yachting could bring positive life changes. In addition, Jang and Lee identified the sense of unity with nature as an important element of satisfaction among respondents to a survey on the yachting subculture [40]. Therefore, we consider it necessary to provide intrinsic facilitation such as by developing programs that can provide a sense of unity with nature, and indeed, these factors are why yachting is in the spotlight even during the constraints imposed by COVID-19.

With the prolonged COVID-19 outbreak, yachts emerged as the blue ocean in the entertainment industry as domestic travel entertainment programs began to turn their eyes to the rare outdoors. For instance, Korea’s MBC began broadcasting the representative examples “Yacht Expedition” and “Sea Road Expedition.” These shows film in the middle of the sea, and in an interview with Seoul Economic Daily, P. D. Park asserted that yachts are a fun hobby because the entry barrier is not high and contact with the surrounding environment is minimal [41]. Owing to the influence of such mass media products, the leisure facilitation of yachting as a nature-based pastime was continuously increasing. Therefore, it was judged that the quality of life could be improved through leisure activities called yachts, and leisure satisfaction can be felt through a sense of unity with nature.

In research on golf participants, Oh found that, as leisure facilitation increased, psychological satisfaction, life satisfaction, and leisure participation also increased, which had positive impacts on leisure satisfaction; Park also found that leisure satisfaction increased as participation in park golf increased [42,43]. Park, Yoon, and Han argued that golf as a leisure activity was a representative wellness sport that enhances leisure satisfaction and that leisure satisfaction through golf participation increases life satisfaction [44]. In short, the findings of the current study support the extant findings of a positive effect of leisure facilitation on leisure satisfaction in terms of interpersonal leisure promotion factors. In the case of golf participants, we judged that leisure facilitation such as fun, competition, human relations, and gaining information not only promote leisure satisfaction but also enables happy lives.

Third, we found that recreation specialization among participants in yachting and golf had a positive effect on leisure satisfaction. Outdoor sports activities such as golf and yachting required consistent engagement, and continuous participation increases recreation specialization; this, in turn, could increase leisure satisfaction driven by stress relief and skill development. Ragheb and Griffith found that the higher the leisure participation among a group of seniors, the higher their leisure satisfaction, and their higher leisure satisfaction had a significant effect on life satisfaction [45].

Researchers also established that marine sports participants experienced pleasure through improving their professional yachting skills as well as through boating-related stress relief and freedom. For instance, Nam and Kwon determined that enjoyment, daily escape, and skill improvement had positive influences on recreation specialization in marine sports [46]. Yachts are recognized as a luxury leisure product in the market, with costs that can act as a constraint to entry in the sport, however, participation is recently increasing owing to increases in the number of aprons along with free education and experience and the low education costs [47].

In terms of golfing, Kim [48] suggested that golf increases happiness, and Yoo reported that recreational specialization has a significant effect on happiness [49]. Kim found that leisure specialization had a static effect on subjective well-being, and Kim, Choi, and Kim found a static effect of recreational specialization on psychological well-being among golfers [50]. Lee, Choi, and Lee also identified that the higher participants’ motivation was to engage in screen golf, the greater their leisure satisfaction and physical well-being [51]. Given that increasing recreation specialization appears to increase leisure satisfaction among golf and yachting participants, efforts are needed to identify the constraints on outdoor sports imposed by the COVID-19 era and to develop strategies for improving leisure facilitation and recreation specialization.

## 5. Practical Applications

Based on the discussion and conclusions above, we propose the following suggestions for follow-up research. First, in this study, we confirmed the correlations among leisure facilitation, recreation specialization, and leisure satisfaction of outdoor sports participants. However, these variables are not the only components or factors that explain outdoor sports; future researchers should review more in-depth, complex causal models that consider related variables such as leisure attitude, leisure constraints, and reentry intention. Second, the responses of the outdoor sports participants in this study could have been influenced by the established background variable of COVID-19, however, we did not consider this potential variable; we consider that further research is needed on whether other variables other than COVID-19 might influence the judgment of outdoor sports participants. It is hoped that research and interest in outdoor sports and leisure will continue through the results of this paper.

## 6. Limitations and Future Research

This study might have the following limitations. First, the study findings have limited generalizability because we recruited the sample from among residents of the Seoul Capital Area (Seoul, Gyeonggi, and Incheon). A follow-up study is needed with findings that can be generalized to the country’s full population. Second, we only investigated golf and yachting, which cannot be considered to represent all outdoor sports; studies are needed on outdoor sports such as mountain climbing, fishing, and skiing. Third, we conducted only a cross-sectional study on the relationships between leisure facilitation, recreation specialization, and leisure satisfaction of outdoor sports participants, but the study’s contributions could be extended by findings from longitudinal studies conducted over a long period of time. Fourth, research is needed on a comparative analysis of outdoor sports before and after the COVID-19 outbreak. Finally, we did not analyze demographic variables such as education level and occupation of outdoor sports participants, and a qualitative study on the nature of outdoor sports enthusiasts’ participation in leisure cultural activities could generate additional interesting research results.

## 7. Conclusions

The aim of the current study was to investigate the relationships among leisure facilitation, recreation specialization, and leisure satisfaction for outdoor sports participants. In particular, we studied golf and yachting participants because these two activities show different patterns from those for many sports during the COVID-19 crisis. The study subjects were 431 adult male and female residents of Seoul, Incheon, Gyeonggi-do, and Chungcheong-do who were participating in golf and yacht activity, and the data processing methods used were frequency analysis, reliability analysis, confirmatory analysis, correlation analysis, and structural equation modeling. We derived the following research results through the above-mentioned research process. Firstly, leisure facilitation did not influence recreation specialization among outdoor sports participants in golf and yachting. Second and third, however, leisure facilitation did positively influence golfers’ and yachters’ leisure satisfaction, and their recreation specialization did influence their leisure satisfaction. These results suggest that it is necessary in the modern era of coronavirus and COVID-19 to better understand the current leisure constraints on outdoor sports and develop leisure facilitation strategies that can overcome them.

## Figures and Tables

**Table 1 ijerph-18-08128-t001:** Characteristics of outdoor sports participants.

Variable		N	%
Gender	Male	228	59.9
	Female	203	47.1
Outdoor sports	Golf	248	57.5
	Yachting	183	42.5
Age	30s and under	79	18.3
	40s	85	19.7
	50s	138	32.0
	60s and over	129	30.0
Participation duration	Under 1 year	98	22.7
	1 year–under 3 years	125	29.0
	3 years–under 5 years	119	27.6
	Over 5 years	89	20.7
Participation intensity	Under 1 h	152	35.3
	1 h–under 3 h	148	34.3
	Over 3 h	131	30.4

**Table 2 ijerph-18-08128-t002:** CFA Fit Indices.

Variables	χ^2^/df	CFI	TLI	RMSEA
Leisure facilitation	2.702	0.946	0.926	0.063
Recreation specialization	2.746	0.964	0.954	0.064
Leisure satisfaction	2.899	0.976	0.960	0.066

**Table 3 ijerph-18-08128-t003:** Results of the Reliability Analysis.

Factor	Sub-Factors	Cronbach’s α	
Leisure facilitation	Intrinsic	0.720	0.858
	Interpersonal	0.743	
	Structural	0.826	
Recreation specialization	Cognitive	0.923	0.964
	Behavioral	0.929	
	Emotional	0.920	
Leisure satisfaction	Stress solution	0.896	0.958
	Health promotion	0.903	
	Technical development	0.818	
	Interpersonal relation	0.792	
	Self-development	0.910	

**Table 4 ijerph-18-08128-t004:** Correlation of leisure facilitation, recreation specialization, and leisure satisfaction.

	1	2	3
1. Leisure facilitation	1		
2. Recreation specialization	0.002	1	
3. Leisure satisfaction	0.286 *	0.217 *	1

Note. * *p* < 0.05.

**Table 5 ijerph-18-08128-t005:** Estimated structural relations coefficients.

Hypothesis	Estimate	S.E.	C.R.
Leisure facilitation→Recreation specialization	0.013	0.052	0.224
Leisure facilitation→Leisure satisfaction	0.099	0.050	1.989 *
Recreation specialization→Leisure satisfaction	0.073	0.048	1.841 *
Model fit: χ^2^ = 47.921, χ^2^/df = 1.198, CFI = 0.997, TLI = 0.997, RMSEA = 0.021			

Note. S.E. = standard error; C.R. = critical ratio; * *p* < 0.05.

## Data Availability

Not applicable.

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
