# Peer review of "Effect of Outdoor Sports Participants on Leisure Facilitation, Recreation Specialization, and Leisure Satisfaction: Yacht and Golf Participants"

_ijerph, 2021, doi:10.3390/ijerph18158128_

Round 1

Reviewer 1 Report

Comments:

This is a relatively complete empirical study. The first thing to note is that there is currently no clear evidence that COVID-19 started in December 2019 in Wuhan, China. Then, there are several areas where I believe the paper should be strengthened.

  1. In abstract, “the mediating effect of the relationship between leisure facilitation and leisure satisfaction”, this sentence is inappropriate. The mediating effect is absent in study hypothesis and results.
  2. The introduction focused too much on the impact of COVID-19 on the sports and leisure industries, but did not survey the impact in this study.
  3. It can be seen from the discussion that there should be differences between activities, because the policies for the two activities are different. So there might be deference between the data of the two samples.
  4. There is a contradiction in the interpretation of the results of H1 and H2 in the discussion. On the one hand, the epidemic has reduced leisure facilitation and is not conducive to leisure specialization. On the other hand, mass media products in the epidemic are used as a leisure facilitation, promoting everyone to participate in outdoor sports.
  5. 3% of the participants in the sample have participated for more than three years, and their participation should not be explained by COVID-19.

Author Response

Reviewer 1

This is a relatively complete empirical study.

The first thing to note is that there is currently no clear evidence that COVID-19 started in December 2019 in Wuhan, China.

Yes, that's right. The COVID-19 sentence has been deleted. Also, the Corona-19 related content has been corrected.

Then, there are several areas where I believe the paper should be strengthened.

In abstract, “the mediating effect of the relationship between leisure facilitation and leisure satisfaction”, this sentence is inappropriate. The mediating effect is absent in study hypothesis and results.

Thanks for the accurate information. Since the mediating effect was not verified, it was corrected.

The introduction focused too much on the impact of COVID-19 on the sports and leisure industries, but did not survey the impact in this study.

Yes, that's right. As the leisure industry develops such as golf and yacht, it was used as data to support the necessity of this study.

It can be seen from the discussion that there should be differences between activities, because the policies for the two activities are different. So there might be deference between the data of the two samples.

The same survey was conducted for participants in outdoor sports such as golf and yacht. The purpose of this study was to analyze the relationship between leisure facilitation, recreational specialization, and leisure satisfaction of golf and yacht participants.

There is a contradiction in the interpretation of the results of H1 and H2 in the discussion. On the one hand, the epidemic has reduced leisure facilitation and is not conducive to leisure specialization. On the other hand, mass media products in the epidemic are used as a leisure facilitation, promoting everyone to participate in outdoor sports.

In research hypothesis H1, the corona pandemic may be considered as the cause of leisure facilitation not reaching recreational specialization. According to the research hypothesis H2, the effect of media can be considered as the cause that leisure facilitation has a positive effect on leisure satisfaction. Although the above hypothesis may be contradictory, it has been described by comparing and analyzing the opinion of the researchers on the research results and previous studies.

3% of the participants in the sample have participated for more than three years, and their participation should not be explained by COVID-19.

The participation period was analyzed as those who participated in golf and yachting activities before Corona.

Reviewer 2 Report

Dear authors,
Overall, the paper is interesting and easy to read. However, I have a few points that caught my eye:
- The sources are largely incorrectly assigned to the digits in the text. A control of the cited literature is only possible with great effort. Some sources are also completely missing in the directory.
- In general: the arguments made are understandable (see discussion ff), but many of the arguments and most of the sources do not seem to be specific to golf or yachting. How do these two outdoor sports differ from jogging, for example? Or to put it another way: Why are these very different sports considered together while others are not? If a clear distinction is not possible, why not investigate outdoor sports that are not characterized by major financial barriers to entry?
- line 55 ff: This development is not related to Covid 19. However, this impression is created in connection with the content of the previous paragraph.
- table 4: In my understanding, the correlation coefficients are small, so the relationships are rather weak. However, this is not discussed in the text?
- line 322: Some sentences are unscientific. Example: "... can lead to positive life changes." This statement is far too vague.

Author Response

Reviewer 2

Dear authors,
Overall, the paper is interesting and easy to read. However, I have a few points that caught my eye:

- The sources are largely incorrectly assigned to the digits in the text. A control of the cited literature is only possible with great effort. Some sources are also completely missing in the directory.

The reference has been fully revised. Thanks for the accurate point.

- In general: the arguments made are understandable (see discussion ff), but many of the arguments and most of the sources do not seem to be specific to golf or yachting. How do these two outdoor sports differ from jogging, for example? Or to put it another way: Why are these very different sports considered together while others are not? If a clear distinction is not possible, why not investigate outdoor sports that are not characterized by major financial barriers to entry?

The reason why we studied golf and yacht participants is that the sports and leisure industry continues to develop. Outdoor sports without financial barriers to entry will be conducted as a follow-up study. Thank you.

- line 55 ff: This development is not related to Covid 19. However, this impression is created in connection with the content of the previous paragraph.

That’s all right, Paragraphs related to Corona have been revised as a whole. Thanks.

- table 4: In my understanding, the correlation coefficients are small, so the relationships are rather weak. However, this is not discussed in the text?

Added interpretation of correlation. Although the correlation coefficient is small, more accurate results can be derived by verifying it with a structural equation model.

- line 322: Some sentences are unscientific. Example: "... can lead to positive life changes." This statement is far too vague.

The above sentence is a translation of previous research. The sentence has been modified so that it is not ambiguous. Thank you.

Round 2

Reviewer 1 Report

This manuscript has been revised a lot, but some places need to be further improved.

  1. In abstract, “the mediating effect of the relationship between leisure facilitation and leisure satisfaction”, this sentence has not been modified.
  2. The introduction focused too much on the impact of COVID-19 on the sports and leisure industries.

Author Response

  1. In abstract, we modify the sentence which it delete mediating.
  2. We reduced the leisure industry and COVID-19. And the introduction is modify that fallow:

    Modern societies have achieved a high degree of industrial growth attributable to technological advances. The changes brought material abundance, and the introduction of a 40-hour work week improved quality of life and increased leisure time and participation in various sports activities [1]. The growth of the outdoor sports market is due to the continuous expansion of numbers of participants [2]. Golfing and yachting are representative untact outdoor sports that share the advantages of using a wide outdoor space and allowing for participation by individuals or small groups, and demand for and participation in both are increasing steadily.

    Following the recent phenomenon of global urbanization, modern people are pursuing nature-based outdoor sports to relieve stressors of urban life such as noise, congestion, and air pollution [3]. According to Statista [4], the world’s largest outdoor sports market, the United States, is worth $36.3 billion, followed by China’s market at $20.5 billion, and the United Kingdom’s at $3.3 billion, with Korea’s market coming in fourth place at $2.1 billion. Given the changes that have taken place already and those predicted for the future, it is necessary to analyze the impacts of COVID-19 on outdoor sports and respond to the changing sports and leisure culture. As of September 2020, COVID-19 had caused sales declines in 7.4% of sports facility businesses, 2.3% of sports equipment businesses, and 2.4% of sports service businesses [5]. The numbers of companies that closed due to cost burdens and other factors increased dramatically.
